# Trunk Water Potential Measured with Microtensiometers for Managing Water Stress in “Gala” Apple Trees

**DOI:** 10.3390/plants12091912

**Published:** 2023-05-08

**Authors:** Luis Gonzalez Nieto, Annika Huber, Rui Gao, Erica Casagrande Biasuz, Lailiang Cheng, Abraham D. Stroock, Alan N. Lakso, Terence L. Robinson

**Affiliations:** 1School of Integrative Plant Sciences, Horticulture Section, Cornell University, Geneva and Ithaca, NY 14456, USA; 2Smith School of Chemical and Biomolecular Engineering, Cornell University, Ithaca, NY 14850, USA; 3FloraPulse Co., Davis, CA 95616, USA

**Keywords:** *Malus* × *domestica*, trunk water potential, water management, microtensiometer, fruit size, fruit growth rate, fruit color, vapor pressure deficit, irrigation scheduling

## Abstract

The weather variations around the world are already having a profound impact on agricultural production. This impacts apple production and the quality of the product. Through agricultural precision, growers attempt to optimize both yield and fruit size and quality. Two experiments were conducted using field-grown “Gala” apple trees in Geneva, NY, USA, in 2021 and 2022. Mature apple trees (*Malus* × *domestica* Borkh. cv. Ultima “Gala”) grafted onto G.11 rootstock planted in 2015 were used for the experiment. Our goal was to establish a relationship between stem water potential (Ψ_trunk_), which was continuously measured using microtensiometers, and the growth rate of apple fruits, measured continuously using dendrometers throughout the growing season. The second objective was to develop thresholds for Ψ_trunk_ to determine when to irrigate apple trees. The economic impacts of different irrigation regimes were evaluated. Three different water regimes were compared (full irrigation, rainfed and rain exclusion to induce water stress). Trees subjected the rain-exclusion treatment were not irrigated during the whole season, except in the spring (April and May; 126 mm in 2021 and 100 mm in 2022); that is, these trees did not receive water during June, July, August and half of September. Trees subjected to the rainfed treatment received only rainwater (515 mm in 2021 and 382 mm in 2022). The fully irrigated trees received rain but were also irrigated by drip irrigation (515 mm in 2021 and 565 mm in 2022). Moreover, all trees received the same amount of water out of season in autumn and winter (245 mm in 2021 and 283 mm in 2022). The microtensiometer sensors detected differences in Ψ_trunk_ among our treatments over the entire growing season. In both years, experimental trees with the same trunk cross-section area (TCSA) were selected (23–25 cm^−2^ TCSA), and crop load was adjusted to 7 fruits·cm^−2^ TCSA in 2021 and 8.5 fruits·cm^−2^ TCSA in 2022. However, the irrigated trees showed the highest fruit growth rates and final fruit weight (157 g and 70 mm), followed by the rainfed only treatment (132 g and 66 mm), while the rain-exclusion treatment had the lowest fruit growth rate and final fruit size (107 g and 61 mm). The hourly fruit shrinking and swelling rate (mm·h^−1^) measured with dendrometers and the hourly Ψ_trunk_ (bar) measured with microtensiometers were correlated. We developed a logistic model to correlate Ψ_trunk_ and fruit growth rate (g·h^−1^), which suggested a critical value of −9.7 bars for Ψ_trunk_, above which there were no negative effects on fruit growth rate due to water stress in the relatively humid conditions of New York State. A support vector machine model and a multiple regression model were developed to predict daytime hourly Ψ_trunk_ with radiation and VPD as input variables. Yield and fruit size were converted to crop value, which showed that managing water stress with irrigation during dry periods improved crop value in the humid climate of New York State.

## 1. Introduction

Climate change is already having a profound impact on agricultural production around the world. Across New York State in the USA, average temperatures are increasing, and precipitation is becoming less frequent. This impacts apple production since fruit size is affected by tree water status [1,2] and the profitability of an apple orchard depends on yield, fruit size and fruit quality. Since apple fruit growth is continuous, maintenance of good water status is needed to reach the maximum size potential at harvest [3,4]. We have introduced a suite of management practices to optimize crop value (yield, fruit size and fruit quality) through precision crop load management [1,5]. The crop load (fruit number) is controlled by precision thinning. However, plant water status throughout the growing season plays an important role in fruit growth and can significantly reduce fruit size and economic value. For example, the severe drought of 2016 in New York State resulted in small fruit size for many apple growers, despite their attempts to irrigate during the summer [6]. The same situation was repeated during the 2022 season, when rainfall during the season was the lowest in the last 5 years.

Precision irrigation is needed to improve the use efficiency of this resource [7]. Automated dendrometers can continuously monitor small fluctuations in fruit and trunk diameters to analyze tree responses to soil water content and atmospheric demands [8]. Fruit dendrometers can be used to detect water stress. Fruit dendrometers have been intensively used in the study of crop management but rarely as water stress indicators [9]. It is necessary to understand fruit growth dynamics in response to physiological and environmental conditions [10]. However, there are no previous studies aimed at understanding the interaction between the dynamics of continuous daily fruit growth and plant water potential (Ψ). The recent introduction of plant microtensiometers (FloraPulse Co., Davis, CA (95616), USA, www.florapulse.com (accessed on 21 December 2022)) allowing continuous measurements of trunk water potential (Ψ_trunk_) provides the opportunity to simultaneously measure trunk water potential and fruit diameter.

Tree water status is an interaction between environmental conditions (primarily vapor pressure deficit), soil water content, light interception, crop and plant development management, irrigation and weeds, and the physiological characteristics of a species [11,12,13]. Currently, soil moisture monitoring, estimates of evapotranspiration from weather data, fruit growth rates, and sap flow measurements are used to estimate plant water needs and schedule irrigation. A precise and direct physiological measurement of the maximum water stress in fruit trees is the stem water potential (Ψ_stem_), which is often measured with a Scholander pressure chamber at midday on leaves enclosed in reflective bags that stop transpiration when there is maximum water stress [14]. As the maximum differences in Ψ_stem_ between irrigation treatments are often detected at midday, this timing has been used by many researchers and some commercial fruit growers to guide irrigation. However, the pressure chamber cannot continuously measure Ψ_stem_ and only provides information on plant water status at the moment of evaluation.

Recently, microtensiometers embedded in the trunk of a tree were developed by Cornell University and FloraPulse Company to continuously monitor Ψ_trunk_ [11]. The first studies with microtensiometers assumed that the sensors measure Ψ_stem_ in the traditional sense of Scholander pressure chamber measurements of leaves. However, Pagay [15] showed that microtensiometers offer yet another plant water status metric. They observed that Ψ_trunk_ measured with microtensiometers embedded in the trunk had a different range of values compared to conventional measures of Ψ_leaf_ and Ψ_stem_ in vineyards. Previous research with these microtensiometers in vineyards [15] and pear [7,16] and apple orchards [11,17] has shown that they give Ψ_trunk_ values very similar to the Ψ_stem_ values obtained with a pressure chamber. However, the distinct advantage is that the microtensiometers can elucidate daily dynamics with continuous measurements without the labor associated with the pressure chamber. Our goal was to establish a relationship between stem water potential (Ψ_trunk_) continuously measured using microtensiometers and the growth rate of apple fruits measured continuously using dendrometers throughout the growing season. The second objective was to characterize a trunk water potential (Ψ_trunk_) threshold to determine when to irrigate apple trees.

## 2. Results

### 2.1. Trunk Water Potential (Ψ_trunk_)

Total seasonal rainfall was higher in 2021 than in 2022. Precipitation was similar in the early part of each season and irrigation was not necessary during April and May (Figure 1). In 2022, the rainfall was higher only in June compared with 2021 (Figure 1). In 2021, rain continued every 2 or 3 days during July and August, and thus irrigation was not necessary during the whole season. However, in 2022, there were two dry periods during the summer. The first two weeks of July and the first two weeks of August required irrigation (Figure 2).

Microtensiometer measurements of Ψ_trunk_ could distinguish between the different water regimes throughout the season. The values of Ψ_trunk_ increased (the trees showed lower water stress) after each rainy period in all three treatments during both seasons, indicating the importance of evaporative demand, with low vapor pressure deficit (VPD) and radiation on rainy days (Figure 2). Moreover, the sensors’ readings were responding to the lower radiation and evaporative demands associated with rainfall because, when precipitation was higher, the measured Ψ_trunk_ was higher. The daily minimum Ψ_trunk_ was lower in the rain-exclusion treatment compared to the other two treatments during both seasons. The rain-exclusion treatment recorded a low of −20 bars for the minimum Ψ_trunk_ in both seasons. Nevertheless, the rainfed and the fully irrigated treatments recorded lows of −14 bars, although the averages for the minimum Ψ_trunk_ were around −8 and −10 bars over the whole season (Figure 2). The microtensiometers showed high sensitivity and could identify the irrigation periods. Thus, the rainfed treatment showed lower Ψ_trunk_ values than the fully irrigated trees during each dry period of the 2022 season (Figure 2).

Using VPD, radiation and the hourly Ψ_trunk_ (rainfed treatment), we constructed MR and SVM models to predict hourly Ψ_trunk_ (Figure 3). Both models used 3889 values for Ψ_trunk_ and the R^2^ values were 0.77 (MR) and 0.78 (SVM). The prediction profiler interactively explains how VPD and radiation factors impact the response of hourly Ψ_trunk_. The prediction profiler in the MR model showed that VPD and radiation were negatively linearly correlated to Ψ_trunk_ (Figure 3). When the radiation and VPD increased, the hourly Ψ_trunk_ declined and the trees showed higher water stress (Figure 3). However, the prediction profiler in the SVM model showed a negative curvilinear relationship between hourly Ψ_trunk_ and either radiation or VPD. Thus, when the hourly Ψ_trunk_ was between −1.77 and −8.73 bars, the profiler showed a negative linear correlation with radiation and VPD. However, when the VPD was greater than 1.5 kPa or the radiation was greater than 2 MJ·m^−2^, hourly Ψ_trunk_ showed a flat response.

### 2.2. Fruit Growth and Fruit Fresh Weights

The fruit diameter and calculated fruit fresh weight were consistently higher in the fully irrigated treatment and then the rainfed treatment and lowest in the rain-exclusion treatment during both seasons (Figure 4). Fruit diameter growth rate was higher during the first few weeks of the season and then gradually declined until harvest in all treatments. This is expected for a one-dimensional measure of three-dimensional fruit. However, the estimated fresh fruit weight showed linear growth throughout the season (Figure 4), indicating that the actual fruit growth rate was quite constant across the whole season in both years. Whether evaluating fruit weight or diameter, there were no significant differences between any of the irrigation regimes until the second week of July. After July, the rain-exclusion treatment showed lower fruit weight and diameter compared to the rainfall-irrigated (rainfed) and fully irrigated trees (Figure 4). The rainfed treatment did not show significantly different fruit diameters compared to the fully irrigated trees during the entire season. However, fruit weight did significantly differ during the last month of the season between these treatments, being lower in the rainfed treatment (Figure 4).

All irrigation treatments showed daily fruit shrinkage in July but not in August or September. Diurnally, the greatest fruit shrinkage was at midday (between 12 h and 14 h) (Figure 5). In July, the fruit shrink started at 10 a.m. in all irrigation treatments and finished at 4 p.m. for the rainfed treatment and the fully irrigated trees and at 6 pm for the rain-exclusion treatment. Irrigation reduced the magnitude and the duration of the shrinkage. During the dry period of July, fruits showed greater shrinkage in the rain-exclusion treatment near midday than in the fully irrigated and rainfed treatments. During August, we did not observe fruit diameter shrinkage, but the swelling rate was lowest at midday, although it stayed slightly positive for all treatments. The rain-exclusion treatment had the lowest fruit swelling rate through most of the day. The daily fruit swelling in September was lower than in July and August and was relatively constant throughout the day, with little midday reduction.

The fruit swelling rate was higher during the nighttime than during the daytime, reaching 0.03 mm·h^−1^ in July and 0.015 mm·h^−1^ in August (Figure 5). The lowest fruit swelling rate occurred with the rain-exclusion trees across the entire 2022 season. The fully irrigated treatment and the rainfed treatment had higher swelling rates and lower shrinking rates than the rain-exclusion treatments in July and August, but they were not different from each other in July. However, in August and September, the fully irrigated treatment had higher fruit swelling rates than the rainfed treatment during the afternoon hours (Figure 5).

### 2.3. Fruit Growth Rate vs. Trunk Water Potential (Ψ_trunk_)

Comparisons of hourly fruit diameters (mm) measured with dendrometers and hourly Ψ_trunk_ (bar) measured with microtensiometers during the 2022 season showed that they generally fluctuated in tandem with each other (Figure 6). The lowest daily Ψ_trunk_ occurred in the mid-to-late afternoon, which generally coincided with the lowest fruit diameters over each 24 h period throughout season. At that time, the greatest differences in Ψ_trunk_ and fruit diameter between the irrigation treatments were detected. On the other hand, during the nighttime, fruits had greater diameters and less negative Ψ_trunk_. Overall, the Ψ_trunk_ and fruit diameters were lower in the rain-exclusion treatment compared to the other treatments during 2022. The fully irrigated treatment showed less negative values for Ψ_trunk_ during the night compared with the rainfed treatment during August, but there were no differences between these two treatments at other times of the season (Figure 6).

Three measures of Ψ_trunk_—the daily minimum value, the daily maximum value and the daily 24 h average—were correlated with fruit growth rate (Figure 7). The regression with the minimum Ψ_trunk_ (midday) showed the lowest R^2^, followed by the regression using the 24 h average Ψ_trunk_, while the regression with the maximum Ψ_trunk_ (nighttime) had the highest R^2^. The minimum Ψ_trunk_ showed a greater range because this was the time of day with the highest water stress. However, the greater fruit growth rates mostly occurred during the nighttime; thus, the maximum Ψ_trunk_ showed a higher correlation coefficient with the fruit growth rate (Figure 7).

Using the hourly fruit weight rate and hourly Ψ_trunk_, we constructed three logistic models to predict the hourly fruit growth rate from Ψ_trunk_ (Figure 8). The model used 2099 data points for the Ψ_trunk_ and fruit growth rate (g·h^−1^) during the 2022 season. The R^2^ was 0.43. However, the important part of this model is the inflexion point of the curve for the rain-exclusion treatment (−7.77 bars) and the lower asymptote of −9.71 bars. Ψ_trunk_ values more negative than the inflection point were associated with significantly lower fruit growth rates. Negative fruit growth rates began at the Ψ_trunk_ level of −9.71 bars. Using these points, we categorized Ψ_trunk_ values higher than −7.77 as indicative of no stress, while values between −7.77 and −9.71 bars were considered indicative of medium stress and values more negative than −9.71 bars as indicative of significant water stress. Based on this analysis, the Ψ_trunk_ value of −9.7 bars could serve as a threshold for irrigation of apple trees based on microtensiometers (Figure 8). The upper asymptotes of the models show that the highest maximum fruit growth rates were obtained for the fully irrigated treatment, followed by the rainfed treatment, with the lowest maximum growth rate demonstrated for the rain-exclusion treatment.

### 2.4. Crop Value

At harvest, there were no significant differences in fruit numbers per tree (200–206 fruits/tree) among the three treatments, indicating that the fruit crop load adjustment undertaken at 10 mm fruit size resulted in the same crop load between trees (Table 1). The number of fruits·tree^−1^ was 229 in 2022 and was 177 in 2021, but this difference was not significant. Average yield was significantly lower in the rain-exclusion treatment (21 kg·tree^−1^ and 48 t·ha^−1^) compared to the rainfed treatment (27 kg·tree^−1^ and 60 t·ha^−1^) and the fully irrigated treatment (32 kg·tree^−1^ and 71 t·ha^−1^). The yield was not significantly different between years. The rain-exclusion treatment had the smallest fruit size (107 g·fruit^−1^) and smallest fruit diameter (61 mm), followed by the rainfed treatment (132 g·fruit^−1^ and 66 mm), while the largest fruit size was from the fully irrigated trees (157 g·fruit^−1^ and 70 mm) (Table 1). The fully irrigated trees showed an advantage of 40 g·fruit^−1^ in fruit size and a 9 mm advantage in diameter compared to the rain-exclusion treatment. Fruit weight and diameter were higher in 2021 than 2022 because the 2022 season was drier than 2021.

Fruits from fully irrigated trees and rainfed trees had higher red blush areas compared to fruits from the rain-exclusion treatment; however, the temperature below the plastic rain-exclusion shelter was around 1.5 °C higher (data not presented). This difference in the temperature could have caused the reduced red blush area (Table 1). The interaction between treatment and year was not significant for any of these parameters.

The calculated crop value based on the data from this experiment and New York industry standard prices (Table 2) showed significant differences between irrigation treatments (Table 3). The first two columns of Table 3 show the crop values when the actual differences in fruit color observed in the experiments were used to calculate packout and crop value. The crop value was highest for the fully irrigated treatment (USD 15.5·tree^−1^ and USD 34,766·ha^−1^), followed by the rainfed treatment (USD 9.7·tree^−1^ and USD 21,633·ha^−1^), while the lowest crop value was with the rain-exclusion treatment (USD 1.9·tree^−1^ and USD 4210·ha^−1^). There was no significant difference in crop value between the years. The last two columns of Table 3 show the estimated crop value when fruit red color was not considered in the calculation of packout and crop value. To exclude the potential influence of the rain-exclusion shelter on fruit red color, we assigned all treatments the same color, with only the irrigation effect on fruit size considered. The results were similar to those when we also considered fruit red color. Thus, the greater effect of our treatments on crop value was due to the effects of rain exclusion and irrigation on fruit size.

## 3. Discussion

Tools for direct and continuous measurement of plant water potential provide an opportunity for precisely applying irrigation water to minimize water stress or to impose a specific water stress [18]. Our work shows that measurements made using the new microtensiometers from FloraPulse in “Gala” apples allowed for continuous monitoring of Ψ_trunk_ for an entire season, and the work was repeatable over two seasons [2]. The sensors detected the different water statuses of three irrigation treatments applied to “Gala” apple trees and concurred with previous observations of pears by Blanco and Kalcsits [7]. Ψ_trunk_ has been shown to be a sensitive indicator of water stress [15,19]. It appears that continuous monitoring of Ψ_trunk_ with the microtensiometers offers a valuable measure of plant water status and can be used for irrigation scheduling [7]. Our results clearly showed that rainy periods and irrigation events were reflected as a relaxing of midday Ψ_trunk_, indicating lower water stress.

VPD and radiation were used to predict hourly Ψ_trunk_ values via MR and SVM models. Backes and Blanke [20] showed a close correlation between water consumption and VPD and, secondly, radiation. Our models suggest a strong correlation between VPD and radiation and Ψ_trunk_, as shown by Blanco and Kalcsits [16] for pears. Lakso et al. [11] showed a faster response of Ψ_trunk_ (evaluated with microtensiometers) to rapid changes in radiation in apple trees. However, our work is the first reporting a method for predicting hourly Ψ_trunk_ with an MR or a SVM model using radiation and VPD. Both showed good fits with these weather parameters and could be used to predict hourly Ψ_trunk_ with our weather conditions. With the MR model, VPD and radiation were negatively and linearly related to Ψ_trunk_. With higher radiation and/or VPD, Ψ_trunk_ was lower. However, our results from this study with an SVM model showed that the correlation between VPD and Ψ_stem_ was not linear, which concurs with the results of De Swaef et al. [21] and Atay et al. [22]. Moreover, both of those studies showed that high VPD values above 2 and 3 kPa did not further reduce stem water potential, which concurs with the results of our SVM model. The MR model has an easy formula, but the SVM model, although more complex, is more realistic. We plan to validate and improve these models for New York State conditions over the next few seasons.

The main objective of our work was to relate fruit size to Ψ_trunk_ to determine if fruit growth could be monitored with microtensiometers and which trunk water potential could be a threshold for triggering irrigation. Although there are many factors that influence fruit size, the number of fruits per tree is the most important factor, with a negative relationship existing between fruit number and size [23,24]. For this reason, we adjusted the numbers of fruits to the same level in our study to reduce possible confounding effects from crop load on fruit size. In addition to crop load, there are two distinct periods of fruit growth during the season: the early-season cell division period of about 6 weeks and the subsequent cell expansion-only period of about 12–14 weeks [3]. Although stress during the early season can affect fruit size by limiting cell division, in the NY State climate, soils are usually saturated in the early season from winter snow and rainfall; therefore, normally, little water stress can occur. Thus, our differential irrigation treatments began after this initial cell division period. Our results clearly showed that water stress during the second part of the season had a significant effect by limiting the attainment of fruit size potential and crop value. The fully irrigated trees showed the highest fruit growth and yield, followed by the rainfed treatment, while the rain-exclusion treatment clearly had the lowest fruit growth and yield. The same results were detected for the daily dynamics of apple fruit growth measured with fruit dendrometers. Therefore, we postulate that minimizing water stress each day throughout the whole season could maximize fruit growth and final fruit size.

Our results showed a declining fruit growth rate (diameter) with the advance of the season. However, when diameter was converted to estimated fruit fresh weight, we found a relatively constant growth rate throughout the season, which was in agreement with the expolinear fruit growth rate model proposed by Lakso et al. [25] and Lakso and Goffinet [3]. They suggested that the best parameter to work with when evaluating fruit growth rate is fruit weight, as the same fruit growth rate measured as diameter represented a larger weight gain early in the season and smaller weight gain late in the season.

Our results suggested that the hourly fruit growth rate (mm·h^−1^) measured with dendrometers and the hourly Ψ_trunk_ (bar) measured with tensiometers showed a good correlation, concurring with previous observations [26]. Naor [27] reported that fruit size at harvest was correlated with daily trunk shrinkage, predawn leaf water potential and midday Ψ_stem_ in apple trees. We found that the lowest diurnal fruit growth rate and lowest Ψ_trunk_ were at midday when the trees were more stressed due to high evapotranspirative demand, concurring with the results of Naor et al. [28]. The situation was reversed during the nighttime, when Ψ_trunk_ and fruit growth rates were higher due to low evapotranspirative demand. The night Ψ_trunk_ showed a better correlation with fruit growth rate than the average 24 h Ψ_trunk_ or Ψ_trunk_ during the day because the fruit swelling rate is higher at night. Previous reports have shown that the diurnal pattern in diameter growth indicates that fruit lose water during the day [29], resulting in a lower fruit growth rate during the daytime. We explored the relationship between fruit growth rate (g·h^−1^) and Ψ_trunk_ and developed a logistic model to relate hourly fruit growth to Ψ_trunk_ (Figure 7). Our results suggested that Ψ_trunk_ values of −7.77 and −9.71 bars were indicative of medium stress and that Ψ_trunk_ values lower than −9.71 bars were indicative of significant stress, resulting in a negative hourly fruit growth rate. Using this threshold and the hourly Ψ_trunk_ model, we could schedule irrigation when the predicted Ψ_trunk_ was below this threshold to improve fruit growth.

Fruit yield per tree in our study was improved by irrigation and related to the volume of water received by the trees [30]. The improvement in fruit yield was due to larger fruit size, since fruit number was essentially constant between treatments. Naor et al. [30] reported that fruit size responds to any deviation from minimum water stress. Supporting this conclusion, our results showed that small differences in Ψ_trunk_ and in the amount of water received between fully irrigated trees and the rainfed treatment resulted in significant differences in fruit diameter and weight.

We observed a significant reduction in fruit color in the rain-exclusion treatment in both seasons. The reduction in red blush area was the result of either water stress from the rain exclusion or the elevated temperature and reduced light below the plastic covering, which was over the trees only during rainy periods. The temperature explanation would support the observations reported by Iglesias et al. [31]. They suggested that a decrease in orchard temperature in the pre-harvest period would improve fruit red color in apples. However, the reduction in red color could also have been due to the water stress. Tao et al. [32] reported a reduction in the red blush area caused by severe water stress after the last stage of fruit cell division. In contrast to our results, Mills et al. [33] reported that red pigmentation increased with greater water stress, while Lopez et al. [34] also indicated that water stress can improve fruit color. Future experiments without a rain-exclusion shelter are needed to clarify the effects of water stress on fruit red color in apple orchards.

Our results also demonstrated a significant economic benefit from irrigation in a humid climate such as that of New York State. Other studies have also evaluated the economic value of irrigation for optimizing crop production in humid regions such as NY State [35,36]. Growers in humid regions are recognizing the economic importance of irrigation [6]. The economic benefit of irrigation was also significant independent of the red color factor, illustrating that irrigation was the most important factor in our experiment affecting fruit size and crop value.

## 4. Materials and Methods

### 4.1. Plant Material and Site

The experiments were performed in an experimental orchard at Cornell’s Agritech Campus in Geneva, New York State, USA (42°51′ N, 77°01′ W), over two consecutive growing seasons (2021 and 2022). Mature apple trees (*Malus* × *domestica* Borkh. cv. Ultima “Gala”) grafted onto G.11 rootstock planted in 2015 were used for the experiment. The rows were oriented north–south, and the trees were trained as tall spindles with a planting density of 2240 trees ha^−1^ (1.22 m × 3.66 m).

The soil type was a Lima loam (calcareous loamy lodgment till derived from limestone, sandstone and shale) with moderate drainage and high water-holding capacity (USDA, https://soilseries.sc.egov.usda.gov/OSD_Docs/L/LIMA.html, (accessed on 1 January 2023)). Trees were irrigated by drip irrigation. Pruning, herbicide and phytosanitary treatments were applied following standards normally used in apple orchards in the region. Meteorological data were collected from a weather station located 0.5 km from the orchard (42°52′35.9″ N 77°01′50.8″ W).

### 4.2. Treatments and Experimental Design

Three different water regimes were compared in this experiment: rain exclusion to induce water stress, rainfed and fully irrigated. The trees in the rain-exclusion treatment were not irrigated throughout the whole season, except in the spring (April and May; (fruit set and cell division period); that is, these trees did not receive water during June, July, August and half of September. The only way to cause water stress in our trees and weather conditions was to reduce all water during all seasons because the soil was saturated with water in much of the summer. The amounts of water in this treatment were 126 mm in 2021 and 100 mm in 2022 (April and May). During each precipitation event, trees were covered with a portable greenhouse to keep water off the trees and soil (Figure 9). Trenches were laid down at the circumference of the tent to drain the water coming off the roof (Figure 9). The rain treatment received only rainwater (515 mm in 2021 and 382 mm in 2022). The fully irrigated treatment received rain but was also irrigated by drip irrigation following the apple-specific irrigation model developed by Cornell (515 mm in 2021 and 565 mm in 2022) [1]. In 2021, natural rainfall between April and September amounted to 515 mm and the irrigation model did not indicate the need for irrigation. Thus, in 2021, the fully irrigated and rainfed treatments were the same. In 2022, natural rainfall between April and September amounted to 382 mm and the irrigation model indicated the need for irrigation at several times during the season. Moreover, all trees received the same amount of water out of season in autumn and winter (245 mm in 2021 and 283 mm in 2022).

In both years, experimental trees were selected with the same trunk cross-section areas (TCSAs; 23–25 cm^−2^ TCSA), and crop load was adjusted to 7 fruits·cm^−2^ TCSA in 2021 and 8.5 fruits·cm^−2^ TCSA in 2022, with hand thinning to 1 fruit·cluster^−1^ when fruits were 10 mm in size. All trees were subject to the same irrigation and natural rainfall regime during the period between fruit set and 10 mm fruit size. The experimental design was a randomized complete block with four replicates of one tree per plot.

### 4.3. Fruit Growth and Fruit Fresh Weights

During both seasons, fruit diameter measurements were recorded manually for 10 fruits·tree^−1^ (40 fruits per treatment). The diameter measurements were recorded every 3–4 days between 2 July and harvest day in 2021, while in 2022, measurements were recorded every 3–4 days in June, every 7 days in July and every 14 days in August and September. Continuous fruit diameter measurements were recorded in 2022 using fruit dendrometers (Winet srl, Italy) from three fruits per tree (12 per treatment) to continuously monitor the fluctuations in fruit diameter every 10 min between 15 July and harvest day in 2022.

Fruit fresh weights were estimated from the diameter measurements using a regression relationship of weight to diameter, FW = 0.0012 × D^2.7655, R^2^ = 0.99 (Lakso, unpublished data). Fruit growth rate, either as diameter increase or weight increase, was calculated as (diameter or weight day 2—diameter or weight day 1)/number of days between measurements.

### 4.4. Trunk Water Potential (Ψ_trunk_)

The microtensiometer sensors (FloraPulse Co., Davis, CA (95616), USA) were embedded into the tree trunks (around 20 cm above the rootstock) of all treatment trees in the study (four sensors per treatment) in both seasons according to the instructions from the FloraPulse company, as described by Lakso et al. [11] and Blanco and Kalcsits [7]. The Ψ_trunk_ measurements were recorded every 10 minutes using a CR6 data logger (Campbell scientific, Inc, Logan, UT (84321), USA).

For our analyses, we used three key points to estimate water stress: the minimum value of Ψ_trunk_ (greatest stress), which was typically in mid-afternoon; the average value of Ψ_trunk_, which was the average for the whole day (24 h); and the maximum value of Ψ_trunk_ (least stress), which was during the night near dawn.

### 4.5. Crop Value

In both seasons, all experimental trees were harvested on the same day during the commercial harvest season for “Gala” apples. Total fruit yield (kg per tree) and fruits per tree were recorded, while fruit diameter (mm), weight (g) and red blush color (%) were measured with a Greefa computer vision packing line machine.

The fruit were assigned a USDA grade based on size and color. The USDA grades based on fruit red blush area were Utility—0–25% red, No. 1—25–40% red, Fancy—40–67% red, X Fancy—67–80% red and XX Fancy—80–100% red. Yield per hectare was estimated from the yield of each experimental tree multiplied by the planting density, and then the estimated packout of each grade was determined for each experimental tree. Prices were assigned to each fruit grade category based on the statewide average prices of “Gala” apples from New York State apple growers (Table 2). Estimated crop value per hectare was derived from the sum of the value of each grade. The price of water was discounted in 2022 in the fully irrigated treatment (USD 0.0022/l). The amount of water used in 2022 was 183 mm, which represented USD 3.964/ha only in the fully irrigated treatment.

### 4.6. Statistical Analysis

The crop load and crop value statistical analysis was performed using ANOVA in SAS 9.4 (SAS Institute Inc., 2009, Cary, NC (27513), USA). Means were separated using Fisher’s LSD tests at *p* < 0.05. The analysis of fruit weight and diameter over the whole season in both years was performed using constrained linear regression.

A multiple regression model (MR) and support vector machine (SVM) model to predict hourly Ψ_trunk_ using vapor pressure deficit (VPD) and weather data for the rainfed treatment were run iteratively. VPD was calculated from temperature and relative humidity. With the MR model, the most complex interaction term with the highest *p* value was deleted from the model before it was run again. This manual backward elimination continued until only significant (*p* < 0.05) terms remained in the model (Milliken and Johnson, 2001). The SVM algorithm uses a group of mathematical functions that are defined as the kernel. This model uses different kernel function algorithms that adapt for better prediction; in our case, of hourly Ψ_trunk_. These functions can be different types (linear, nonlinear, polynomial, radial basis function and sigmoid). The formula for this model is complex, and the full formula for the best model is not included in this article. The objective of this modeling effort was to predict Ψ_trunk_ using VPD and weather parameters. The weather parameters evaluated in the model were VPD, temperature, radiation, precipitation and relative humidity. VPD and radiation were selected using stepwise multiple regression modeling to predict hourly Ψ_trunk_ values for apple crops.

A logistic model was used to relate fruit growth rate (shrinking and swelling rate (g·h^−1^)) to the hourly Ψ_trunk_ in each irrigation treatment:(1)$$Fruit\;growth\;rate=\frac{c}{\left(1+e^{\left(-a\times \left(\Psi_{\mathrm{trunk}}-b\right)\right)}\right)}$$
where *a* is the growth rate of the asymptotic growth curve, *b* is the inflexion point and *c* is the asymptote.

The linear regression, MR model, SVM model and logistic model were analyzed using JMP16 Pro statistical analysis software (SAS Institute Inc., 2009, Cary, NC (27513), USA).

## 5. Conclusions

The use of microtensiometers and dendrometers allows continuous measurements of Ψ_trunk_ and fruit growth in “Gala” apples. Fruit growth rate and weight increases were influenced daily by tree water status, as measured with the microtensiometers. The hourly fruit growth rate (g·h^−1^) and hourly Ψ_trunk_ (bar) showed a significant correlation. The lowest fruit growth rate and Ψ_trunk_ were at midday, while both reached the highest values during nighttime. Night Ψ_trunk_ showed a better correlation with fruit growth rate than the 24 h average Ψ_trunk_ or the lowest Ψ_trunk_ measured during the midday period. A logistic model developed from this study suggested that the critical value of Ψ_trunk_ was −9.7 bars in New York State. This value could be used as a threshold for starting irrigation of apple orchards. This could be combined with the results from SVM and MR models using radiation and VPD to predict daytime hourly Ψ_trunk_ to automate irrigation in the future. Our results show that irrigation has a significant impact on crop value, even in a humid climate such as in NY State.

## Figures and Tables

**Figure 1 plants-12-01912-f001:**
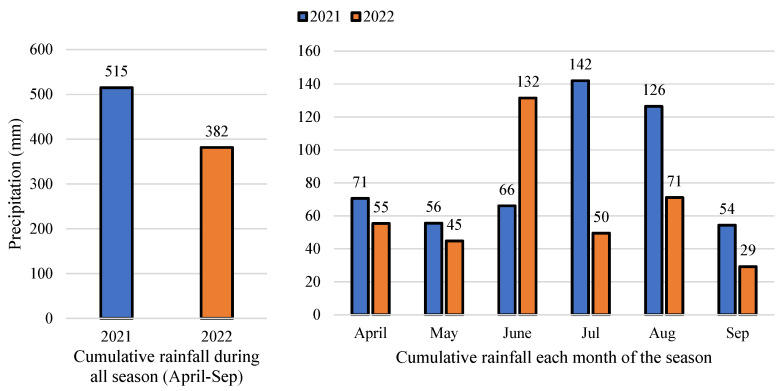
Cumulative rainfall during each month and all season (April to September) in 2021 and 2022 (Geneva, NY, USA).

**Figure 2 plants-12-01912-f002:**
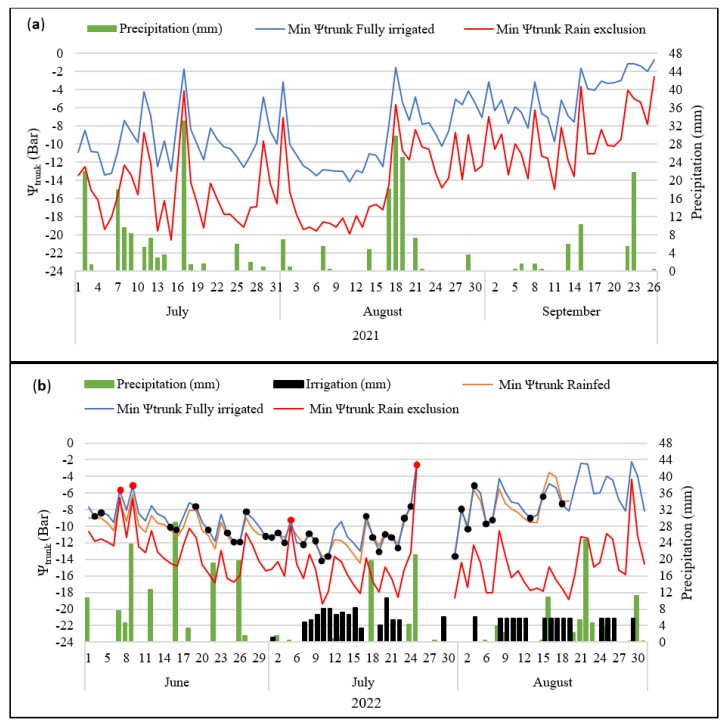
The green line (fully irrigated), yellow line (irrigated with rain only: rainfed) and red line (rain exclusion) are the Ψ_trunk_ (bar) readings throughout the seasons 2021 (**a**) and 2022 (**b**). The minimum Ψ_trunk_ values are the averages from four trees in each treatment at midday. Vertical blue bars along the *x* axis are precipitation amounts (mm) and black bars are the irrigation amounts (mm) during the seasons 2021 (**a**) and 2022 (**b**). Fisher’s LSD tests at *p* < 0.05 resulted in 0.215 bars in 2021 and 0.74 bars in 2022. Red dots denote no significant differences between all regimes of irrigation. Black dots denote no significant differences between fully irrigated and rainfed trees.

**Figure 3 plants-12-01912-f003:**
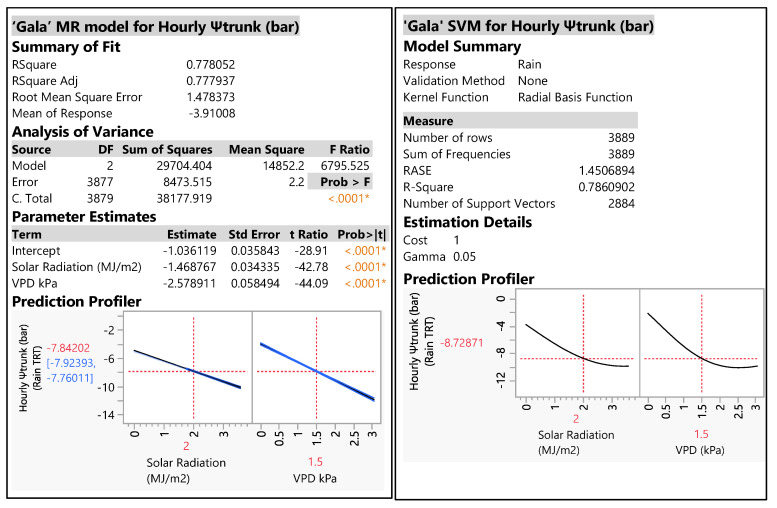
MR model (left) and SVM model (right) built to predict hourly Ψ_trunk_ (bar) during both seasons of study. Summary of fit, analysis of variance, parameter estimates and prediction profiler for “Gala” Ψ_trunk_ (bar). The models used the rainfed treatment. * indicates that one variable was significant.

**Figure 4 plants-12-01912-f004:**
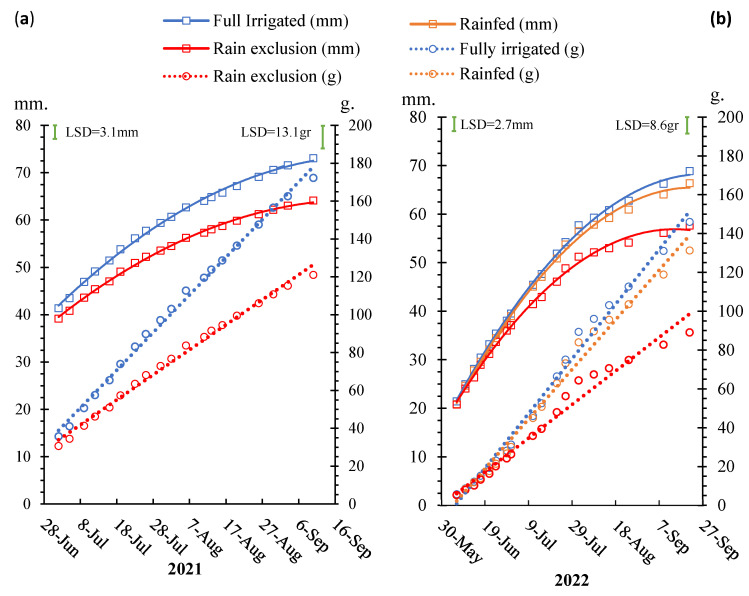
Fruit diameter (mm) and calculated fruit fresh weight (gr) during the 2021 (**a**) and 2022 (**b**) seasons for “Gala” fruit grown under three water regimes. The fruit diameters were analyzed using quadratic regression and fruit weight was analyzed with linear regression (R^2^ = 0.98/0.99 and *p* < 0.0001 for all curves). Fisher’s LSD tests at *p* < 0.05 resulted in 3.1 mm and 13.1 g in 2021 and 2.7 mm and 8.6 g in 2022.

**Figure 5 plants-12-01912-f005:**
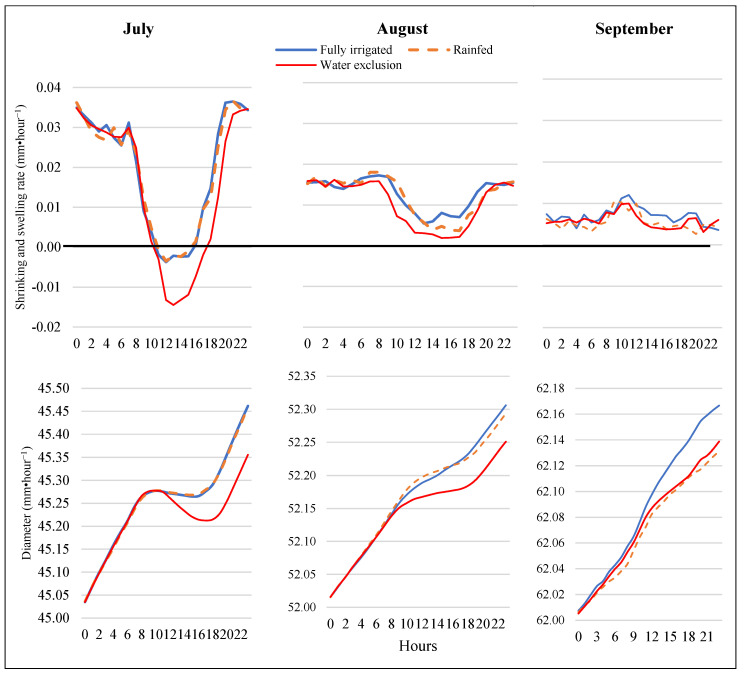
The top figures are the monthly average hourly rates of fruit shrinking and swelling (mm·h^−1^) and the bottom figures the monthly average hourly fruit shrinking and swelling (mm·h^−1^) measured with dendrometers for “Gala” apples under three water regimes during the 2022 season. The values for fruit shrinking and swelling were the averages from all measured fruits per treatment over the entire month. The diameter of fruit at 0 h in the bottom figures was the fruit diameter average from all treatments on the first day of each month.

**Figure 6 plants-12-01912-f006:**
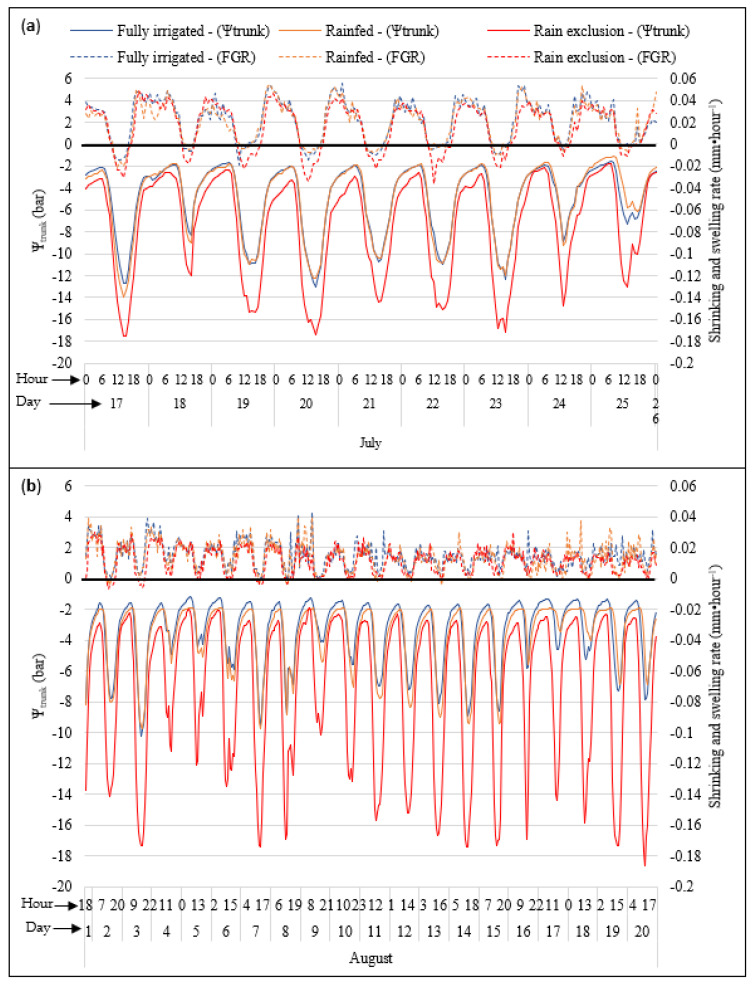
Hourly shrinking and swelling (mm·h^−1^) measured with dendrometers and hourly Ψ_trunk_ (bar) measured with tensiometers during the 2022 season for “Gala” under three water regimes: (**a**) July and (**b**) August.

**Figure 7 plants-12-01912-f007:**
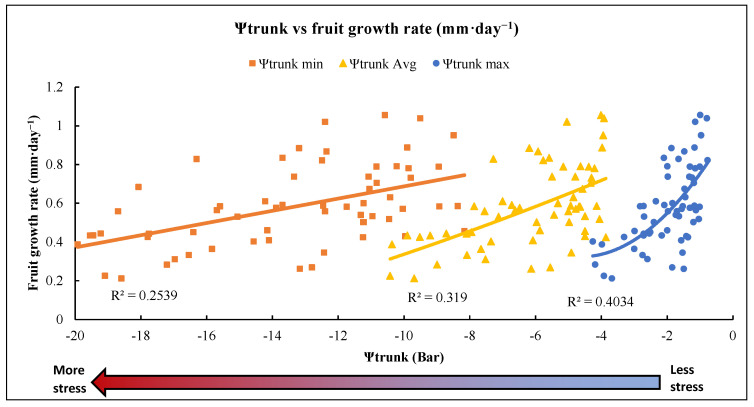
Relationships between Ψ_trunk_ (bar) and daily fruit growth rate (mm per day) during both seasons in a “Gala” apple orchard in Geneva, NY, USA. Min Ψ_trunk_ is the lowest Ψ_trunk_ value around midday (higher stress), Ψ_trunk_ avg is the average for the whole day (24 h) and Max Ψ_trunk_ is the higher Ψ_trunk_ during the night (lower stress).

**Figure 8 plants-12-01912-f008:**
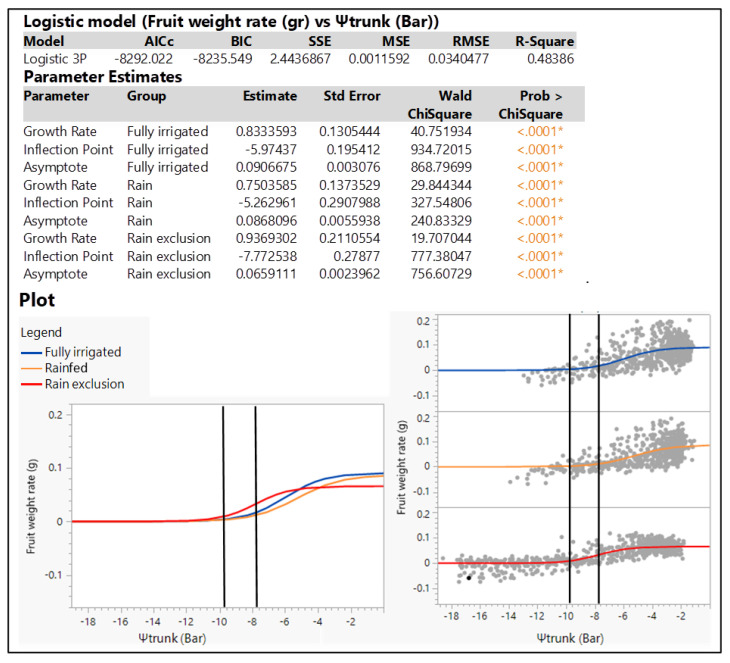
Summary of fit, parameter estimates, and prediction profiler for “Gala” Ψ_trunk_ (bar) models built to predict hourly fruit growth rate (g) during the 2022 seasons of study. Black bars indicate the inflexion point of the curve for the rain-exclusion treatment and the lower asymptote. * indicates that variable calculated for the model was significant.

**Figure 9 plants-12-01912-f009:**
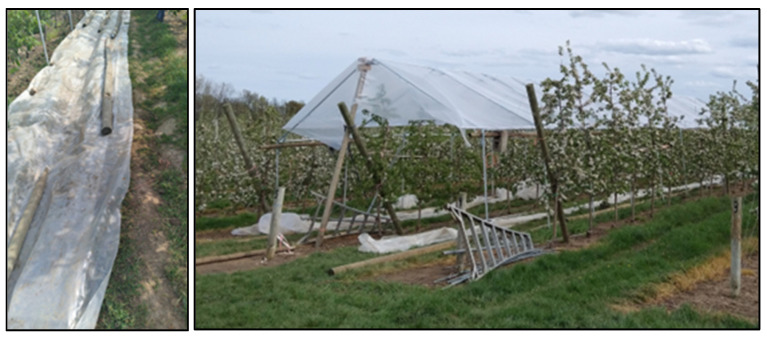
Portable greenhouse installed in the experiment.

**Table 1 plants-12-01912-t001:** Number of fruits·tree^−1^, yield (kg·tree^−1^), tons·hectare^−1^, fruit diameter, fruit weight and red blush area at harvest time for “Gala” apple trees grown under three water regimes.

	Fruits per Tree	Yield (kg/Tree)		Tons/ha		Weight (g)		Size (mm)		Red (%)	
Fully irrigated	206	31.8	a ^*^	71.2	a	157	a	70	a	62	ab
Rainfed	200	26.5	ab	59.5	ab	132	b	66	b	67	a
Rain exclusion	205	21.3	b	47.8	b	107	c	61	c	39	b
**Treatment (Trt)**	**ns**	**<0.01**		**<0.01**		**<0.01**		**<0.01**		**0.108**	
2021	177	25.7		57.7		146	a	68	a	49	
2022	229	27.3		61.1		120	b	64	b	57	
**Year (Y)**	**ns**	**ns**		**ns**		**0.02**		**0.01**		**ns**	
**Trt × Y**	**ns**	**ns**		**ns**		**ns**		**ns**		**ns**	

* Different letters denote significant differences (Duncan’s range test at *p* < 0.05). ns—not significant at *p* < 0.05.

**Table 2 plants-12-01912-t002:** Grower returns (USD/kg) for “Gala”, color category and fruit size category after subtracting storage and packing charges. These included packing charges and the average cost of regular and CA storage. Values were taken from statewide averages for the New York State apple industry.

Grower Returns (USD/kg)	Fruit Size (g)								
Color Category	<128	128 < 136	136 < 153	153 < 167	167 < 190	190 < 215	215 < 238	238 < 264	≥264
XX Fancy	−0.01	0.18	0.84	0.84	0.84	0.95	1.12	1.17	1.17
X Fancy	−0.01	0.18	0.73	0.73	0.73	0.84	1.01	1.06	1.06
Fancy	−0.01	0.18	0.57	0.57	0.57	0.68	0.84	0.90	0.90
No. 1	−0.01	0.18	0.18	0.18	0.40	0.51	0.68	0.73	0.73
Utility	−0.01	0.18	0.18	0.18	0.18	0.18	0.18	0.18	0.18
Grower returns (USD/kg)	Fruit Size (g) (Analysis of All Treatments with the Same Color)								
XX Fancy	−0.01	0.18	0.84	0.84	0.84	0.95	1.12	1.17	1.17

**Table 3 plants-12-01912-t003:** Crop value (USD) per tree and per hectare for “Gala” apple trees grown under three water regimes. The last two columns were analyzed considering the maximum color in “Gala” fruit, assuming all irrigation treatments had the same color.

	USD/Tree		USD/ha		USD/Tree (Same Color)		USD/ha (Same Color)	
Fully irrigated	15.5	a ^*^	34,766	a	18.3	a	41,009	a
Rainfed	9.7	b	21,633	b	10.75	b	24,089	b
Rain exclusion	1.9	c	4210	c	3.44	c	7701	c
**Treatment (Trt)**	**<0.01**		**<0.01**		**<0.01**		**<0.01**	
2021	10.52		23,570		13.4	a	30,046	a
2022	7.39		16,575		8.6	b	19,208	b
**Year (Y)**	**ns**		**ns**		**<0.01**		**<0.01**	
**Trt × Y**	**ns**		**ns**		**ns**		**ns**	

* Different letters denote significant differences (Duncan’s range test at *p* < 0.05). ns—not significant at *p* < 0.05.

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
