# Peer review of "Trunk Water Potential Measured with Microtensiometers for Managing Water Stress in “Gala” Apple Trees"

_plants, 2023, doi:10.3390/plants12091912_

Round 1

Reviewer 1 Report

Dear authors,

This is an interesting and well written manuscript. I really enjoyed reading this research but there are some important gaps that not allowed to be accepted in this form.

Results

Figure 2: Please, note that in Figure 2 (a) 2021 is ablsence and also some lines are not explained.

There is enough the explanation of the parameters once time.

Figure 5: The same probleme with the lines. Why the line of fully irrigated is under the month July, the line Rain under the monthe August and so on? Please, put these lines all together. 

Figure 6: On the x-axis there is the Month; Day; hour words. It is not clear. Please, remake the figure 6 again.

Figure 7. Why the correlations were made only for the year 2022? What happend in 2021?

Figure 8. It is not correct. It is not real! Why the Last two columns were analyzed considering the maximum color in gala assuming all 2 irrigation treatments had the same color? One of the major qualitative differences on these kind of experients (i.e. different irrigation system) is the different color of the fruits.

On the results could be a lot of interesting have an idea how much water was consumed from each treatment and how much biomass produced by each 

Figure 8. The leggend is not clear.

 Discussion

Also in this section there are some parts that are not supported by the results

For example: On lines 348-348 Fruit yield per tree in our study was improved by irrigation and was related to the volume of water received by the trees. There is no reference on the results.

Please, check all the discussion for similar errors.

 Materials and Methods

4.1. Plant material and site

Line 385: There is too space. Please, check it! The same error on lines 394 and 396.

4.2. Treatments and experimental design.

To my opinion, the whole setting of the experiment is wrong. The drain water coming off the roof was brought away or remained to the place. In the second case there is a propably interaction with the rain water that remains on the place. Moreover, there is the use of a portable green house. Are we sure that this kind of plastic used for the portable green house did not modify the fruit caracteristics on this tratment?

Author Response

Thanks for your review, i accept all suggestions and answer your questions in the document attached

Best regards and thanks

Luis

Reviewer 2 Report

I have little concerns about the appropiatness of this ms, other than a better explanation of rain exclusion treatment, however, I have made numerous suggestions to make more unfirm and readable the manuscript. I have marked in yellow too what I think error in the way authors cire refernces and other aspects. I ask the authors to carefully read my comments and suggestions and improve the quality of the presentation.

Author Response

I really appreciate your review. Your revision is very clear. I accept practically all changes that you suggest.
I attach the answer in the pdf document.
Thank you very much for your review

Best regard

Round 2

Reviewer 1 Report

The authors support that the the experiment is essential for the growers learn to manage the irrigation.

If it so, why they do not report how much irrigation water apple trees need to produce the yields and the fruit sizes from each irrigation treatment???

Also, in Table 2 is reported the Crop value ($USD) per tree and per hectare on ‘Gala’ apple trees grown under 3 water regimes. This table has no sense if a consideration of the irrigation cost is not made.

Author Response

Thanks for all your suggestions

I answer down

The authors support that the the experiment is essential for the growers learn to manage the irrigation.

If it so, why they do not report how much irrigation water apple trees need to produce the yields and the fruit sizes from each irrigation treatment???

I put the amounts of water in each treatment in Materials of methods in the point 4.2.
Thanks for this comment I found one mistake in the figure 1 and I modified. I had a mistake in the column that I used to generate this graph.

Also, in Table 2 is reported the Crop value ($USD) per tree and per hectare on ‘Gala’ apple trees grown under 3 water regimes. This table has no sense if a consideration of the irrigation cost is not made.

I Modified my analisis and discounted the price of water. I only discounted in fully irrigated trees in 2022 because in 2021 we never turned on the irrigation because the raining was sufficient.
You can see en materials of methods all information
The result the analisis is the same. 

Reviewer 2 Report

Line 16. I repeat the suggestion: the treatment no rain has to be explained clearly. If, as the a first author indicates is no rain and no irrigation (what is hardly possible commercially), authors should add that the treatment was applied only during a short period indicating dates of beginning and end, and informing that previous winter rain could have replenished water soil content. This has to be clearly explained her and in M&M too.

Line 20. I insist here too. I need to know fruit load and comment if this is an off year or not. I don’t understand the answer of the authors: “Its strange years off”

Line 39. Crop load needs a reference of the size of the tree. This can be done as fruit per canopy volume, fruit per TCSA or whatever, but the same fruit number does not mean the same for a small young tree than for a huge adult tree

Line 62. Again insufficient. When I stated that a crop includes vegetation. And ask if the authors mean fruit and vegetative growth? He responds to a different question related to cover crops and/or weeds. Please, rephrase the whole sentence. We don’t use the term vegetation for cover crops.

L104 I do not understand author’ explanation.

L123. To my suggestion of using rainfed for trees receiving water only from rain, the author answered: “I think that rainfed is no correct because the trees was irrigated by rain”. Unless the farmers store rain water in a reservoir and apply as irrigation water, authors is very wrong. It is not correct assume that rain is irrigation. This is not true for many reasons. Please, think and check literature, please. And modify accordingly.

L154. I wait for an answer as I do for my question of Line 390 and 432 and 480 and so on. I suggest author to give response point by point to all my suggestions. I cannot follow if many of the suggestions were taken into account because he did not give explanations to them.

Please, take your time for the corrections and give detailed answers

Author Response

Thanks for all your suggestions

I answer down and in the PDF

Line 16. I repeat the suggestion: the treatment no rain has to be explained clearly. If, as the a first author indicates is no rain and no irrigation (what is hardly possible commercially), authors should add that the treatment was applied only during a short period indicating dates of beginning and end, and informing that previous winter rain could have replenished water soil content. This has to be clearly explained her and in M&M too.

I add this information in the abstract and materials of methods. I hope that now is weel

Line 20. I insist here too. I need to know fruit load and comment if this is an off year or not. I don’t understand the answer of the authors: “Its strange years off”

Gala rarely has years off in a commercial orchards. We have one project to evaluate the effect of the crop load in the return bloom and others parameters and i only can see effects in the return blooms in crop loads the 20 fruits per TSA in trees with 15 cm2 TSA. However, These trees with 20 fruits/TSA showed good flowering. Its different in other cultivars, Honeycrisp, Fuji, Golden.. These cultivars have high alternance of yield but Gala no.

Line 39. Crop load needs a reference of the size of the tree. This can be done as fruit per canopy volume, fruit per TCSA or whatever, but the same fruit number does not mean the same for a small young tree than for a huge adult tree

I add more specific information of TSA in the abstract and material and method. In the last version of the manuscript in materials of method was explained this information

  In both years, the experimental trees were selected with the same trunk cross-section area (TCSA) and crop load was adjusted to 7 fruits•cm-2 TCSA 

Line 62. Again insufficient. When I stated that a crop includes vegetation. And ask if the authors mean fruit and vegetative growth? He responds to a different question related to cover crops and/or weeds. Please, rephrase the whole sentence. We don’t use the term vegetation for cover crops.

This sentence is a reference the other paper, i put in the same order that Alan Lakso put in his paper. I change the expression. I hope that you consider that now is better. When i answer about the weed i though that you talk in your comment the the expression competing "floor management" that i changed for weeds. Im sorry for the confusion

L104 I do not understand author’ explanation.

In the first version of this the paper i justify that  Ψtrunk increase for the for the soil moisture and in the review the Alan (Author 7) he put this comment (indicating the importance of evaporative demand, not just soil moisture). When i modify all changes i don't modify this sentence. In the last revision, I modified this sentence. I'm sorry

L123. To my suggestion of using rainfed for trees receiving water only from rain, the author answered: “I think that rainfed is no correct because the trees was irrigated by rain”. Unless the farmers store rain water in a reservoir and apply as irrigation water, authors is very wrong. It is not correct assume that rain is irrigation. This is not true for many reasons. Please, think and check literature, please. And modify accordingly.

Im sorry for the confusion, I change the name in this revision. When i said this i only want say that trees that receive 300-500 mm of rain during the season are not in situation of rainfed. However i accept you sugestion.

L154. I wait for an answer as I do for my question of Line 390 and 432 and 480 and so on. I suggest author to give response point by point to all my suggestions. I cannot follow if many of the suggestions were taken into account because he did not give explanations to them.

I ansew in the PDF

Please, take your time for the corrections and give detailed answers

Round 3

Reviewer 1 Report

No more comments.